# How Far Does the European Union Reach? Foreign Land Acquisitions and the Boundaries of Political Communities

**Torsten Menge** 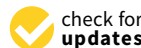

Liberal Arts Program, Northwestern University in Qatar, Education City, P.O. Box 34102, Doha, Qatar; torsten.menge@northwestern.edu

**Abstract:** The recent global surge in large-scale foreign land acquisitions marks a radical transformation of the global economic and political landscape. Since land that attracts capital often becomes the site of expulsions and displacement, it also leads to new forms of migration. In this paper, I explore this connection from the perspective of a political philosopher. I argue that changes in global land governance unsettle the congruence of political community and bounded territory that we often take for granted. As a case study, I discuss the European Union's Renewable Energy Directive as a significant driver of foreign land acquisitions. Using its global power, the European Union (EU) is effectively governing land far outside of its international borders and with it the people who live on this land or are expelled from it. As a result, EU citizens ought to consider such people fellow members of their political community. This has implications for normative debates about immigration and, in particular, for arguments that appeal to collective self-determination to justify a right of political communities to exclude newcomers. The political community to which EU citizens belong reaches far beyond the EU's official borders.

**Keywords:** collective self-determination; global land rush; land governance; large-scale land acquisitions; legitimacy; migration; political community; territory; transnational governance

## 1. Introduction

The world-wide surge in large-scale foreign land acquisitions at the beginning of the 21st century—often referred to as a "global land rush"—marks a radical transformation of the global economic and political landscape. This includes an increasing financialization of the global economy, changes in the organization of agricultural production and trade, and a shift, in many parts of the world, towards forms of land governance that focus on the flow of resources and goods [1,2]. This surge in land acquisitions is also linked to migration, as land that attracts capital often becomes the site of expulsions and displacement. At the same time, European countries and the United States—significant sources of foreign capital—have adopted restrictive immigration policies and fortified their borders [3].

In this paper, I analyze the connection between foreign land acquisitions and migration from the perspective of a political philosopher. I argue that changes in global land governance unsettle the congruence of political community and bounded territory that we often take for granted. As a case study, I discuss the role of the European Union's Renewable Energy Directive as a significant driver of foreign land acquisitions. Using its global power, the European Union (EU) is effectively governing land far outside of its international borders, and with it the people who live on this land or are expelled from it. I argue that as a consequence, EU citizens should consider such people fellow members of their political community. In other words, the political community to which EU citizens belong reaches far beyond the EU's official borders.

The current pattern of transnational land acquisitions has so far received little attention from political philosophers and political theorists (with a few exceptions, e.g., [4,5]). This is unfortunate, as it raises complex questions about global justice, environmental justice, global governance, migration, and so on. One explanation for this lack of attention is that many political philosophers and theorists continue to work within the framework of *methodological nationalism*, taking the territorially bounded nation-state as their descriptive and normative starting point [6,7]. Within the social sciences, this framework has been criticized for its descriptive and explanatory inadequacy. My focus here will be on its normative adequacy. Within normative debates, methodological nationalism is reflected in the assumption that political communities usually overlap with territorially bounded nation-states. Based on this starting point, some political philosophers have argued that only people who reside within a state's territory have standing to participate in the community's collective decision-making, and that such community has a (qualified) right to exclude newcomers from its territory. I will argue that arguments that appeal to such views come under pressure once we consider the increasing territorial reach of wealthy Western countries.

The increase in large-scale foreign land deals involves complex and polycentric relationships that cannot be reduced to simple North-South cleavages [8,9]. My case study about the role of the European Union only addresses one aspect of this development. I approach the topic from this perspective because—being an EU citizen myself—I want to consider its implications for the political self-understanding of EU citizens. The claims I make here about the nature and legitimate boundaries of political communities should be understood within that context. I do not intend to make general claims about the normative implications of foreign land acquisitions for any political community. I suspect that there are important moral asymmetries, though I cannot discuss them in this paper. It may well be the case, for example, that the citizens of powerful countries (especially former colonial powers) have a political responsibility towards many people outside of and at their borders, while people whose livelihoods are threatened by large-scale land deals are entitled to reject many political and economic connections because they result from and reinforce domination. In this paper, I will focus only on the former issue.

In the following, I first introduce a number of philosophical questions about the legitimate boundaries of political communities (Section 2). I then provide a brief overview of current patterns of large-scale foreign land acquisitions, focusing in particular on their effects on global land governance (Section 3). I discuss the EU's Renewable Energy Directive to illustrate some of these effects, arguing that the EU effectively governs land outside of its official borders and with it the livelihoods of those inhabiting it (Section 4). To elaborate the argument that this has implications for the legitimate boundaries of the political community to which EU citizens belong, I build on Iris M. Young's account of political responsibility (Section 5). I defend the argument against the objection that the social connections created by foreign land acquisitions are not sufficient to give rise to political obligations (Section 6), before ending with a short conclusion.

## 2. The Legitimate Boundaries of Political Communities

The concept of *political community* is foundational to political theory and political philosophy. The term 'political community' draws attention to the special relationships among the members of a nation, a people, the *demos*, a country, and so on, relationships which arguably ground the collective authority of such a community to govern its shared life. When we identify a group as a political community, we implicitly make claims about who has the right to make collectively binding decisions, to whom these decisions apply, and who ought to have a say in making them. Political community is thus at its core a normative notion. This is not to deny that social scientists and others can use the concept to identify and describe groups that make and implement decisions that are *taken to be* collectively binding. But this descriptive use does not settle the normative questions that arise when we ask about the legitimacy of their authority.

It is widely assumed that political communities have a right to self-determination. That is, members of a community have the right to collectively determine the character of their shared life together without outside interference or domination by others. This is an important principle in international law, though how it is properly interpreted is a matter of some debate [10]. As a moral principle, it plays a central role in a number of debates within political philosophy. In the global justice literature, for example, self-determination is often considered to be an important value that may be in tension with other considerations, such as the protection of human rights or global distributive justice. Some philosophers have defended the idea that significant inequalities between states can be a legitimate outcome of collective self-determination [11–13]. In debates about immigration, some political philosophers have argued that collective self-determination grounds a qualified right of states to exclude would-be immigrants (see [14,15] for an overview of the debate). Proponents of this kind of argument believe that it provides a distinctly liberal justification for a right to exclude because the commitment to self-determination is an expression of the foundational liberal idea that legitimate political authority derives from the people [14].

Consider, for example, the contributions of the philosopher David Miller to these debates [16]. Miller has argued that the preservation of a distinctive national culture is a legitimate role of nation-states. A national culture, a set of overlapping beliefs, practices, and sensibilities, is important for the sense of identity and belonging of its members, and it provides them with a basis for making more individual choices. People thus have a legitimate interest in controlling the public culture of their country by, among other things, regulating immigration [11]. He rejects arguments for migration that appeal to a demand for global equality of opportunity, arguing that the scope of distributive justice is set by bounded political communities. Miller does recognize some ethical constraints on immigration policies: He argues that states have certain duties towards refugees, and he rejects the use of racial, ethnic, or cultural criteria for selecting immigrations. In these cases, the interest of political communities in determining their own character has to be weighed against the interest of migrants. However, collective self-determination carries significant weight, allowing states to decide how many immigrants to admit or to close their borders to new immigrants altogether.

Theorists are not the only ones appealing to the principle of self-determination. In the ongoing public debates about migration in the EU, for example, political actors often portray attempts to exclude migrants as a legitimate exercise of state sovereignty. Indeed, while the issue is philosophically controversial, it is usually taken for granted in public debates that states have a right to exclude would-be immigrants. Fear of cultural change and loss of control are also connected to claims to self-determination, though such appeals are often more illiberal than liberal (see for example [17,18]).

But while the right to collective self-determination is relatively uncontroversial, its application is anything but straightforward. To make this appeal in a concrete case, we need to know who the "self" of self-determination is. For example, if we contend that European countries have this right and consequently have discretion over whom to admit as new members, we need to know who legitimately belongs to the community and is entitled to participate in its decision-making. Answering this question in a non-arbitrary fashion is surprisingly difficult. Since membership questions are central to the character of a community, they can arguably be decided only by the community itself. But for the community to make that decision, we first need to determine who can legitimately participate in making it. And that determination, it seems, can in turn be made only by the community itself. This means that we need to start by determining who is a legitimate member of the community, and so on. The question who legitimately belongs to a political community thus generates an infinite regress, a problem that political theorists refer to as the *boundary problem* [19,20].

The regress is a problem because we seem to be able to stop it only by taking some historically contingent boundary as given. That is unsatisfactory because political communities claim special authority to govern the shared lives of their members, to use coercive force, and to exclude non-members from some opportunities and social goods. Drawing the civic boundaries of a community in one way rather than another is therefore in need of moral justification, and merely contingent

boundaries do not seem to provide it. Some political theorists have drawn the conclusion that the *demos* is in principle unbounded [20]. This claim is not intended as a blueprint for institutional reality, but it would imply at least that potential outsiders have a right to participate in deciding how political borders ought to be drawn. From this perspective, it is an open question whether the citizens of EU countries can legitimately treat migrant as outsiders whom they have a right to exclude. Many theorists, in order to stop the regress, appeal to an All Affected Principle, which holds that all who are affected by a decision have a right to participate in making it, or an All Subjected Principle, which holds that all and only those who are subject to a state's laws are entitled to political participation. However, it is contested what these abstract principles imply when it comes to determining concrete borders [21,22].

The boundary problem is a tricky theoretical challenge, but in practice, most political communities we recognize are organized in the form of territorially bounded nation-states [23]. The appeal to territory stops the regress by answering the question 'Who is the people?' with: everyone who resides in this territory. It takes for granted territorially bounded nation-states as the starting point for political and moral analysis, restricting political participation typically to those who reside within a bounded territory (though this leaves open whether *all* long-term residents have such rights [24]). Appealing to territorial boundaries might appear to be morally inadequate because it seems to construe territory as an objective, pre-political starting point, ignoring the fact that territorial boundaries are themselves the effect of political struggles [23]. But maybe the appeal is not completely irrelevant. Territory could be seen as a proxy for identifying groups of people whose interests are closely intertwined, as their actions mutually affect one another more directly than they affect those who live further away [21,25]. A territory is not simply an abstract location; it is part of the material infrastructure that makes the distinctive life of a group possible. Indeed, some philosophers have argued that a bounded territory is necessary for political self-determination [26,27]. Insofar as there is a geographical overlap between shared political institutions, shared social and economic practices, and a shared material infrastructure, it may well be reasonable to identify the boundaries of a political community by appeal to territory.

Whether there is such an overlap is an empirical question, and in an increasingly globalizing world we have reason to deny it. Linda Bosniak, for example, has suggested that "[t]erritoriality as an ethical concept is becoming less authoritative, largely because territory itself no longer organizes social and political life in the determinative way it once did" [28] (p. 490). The recent surge in large-scale land acquisitions provides a useful case study to elaborate Bosniak's claim. While it is very much an evolving story and its scale, character, and impact are contested, the extensive literature about it provides a sufficient basis for assessing assumptions about the normative import of territory for questions about immigration, global justice, and transnational governance.

## 3. Large-Scale Foreign Land Acquisitions and Changes in Land Governance

While there is disagreement about the precise scale of the recent surge in large-scale foreign land acquisitions that started in the early 2000s, due to a lack of reliable empirical data, most authors agree that it reflects profound economic and social transformations [29,30]. The surge is often referred to as a "global land rush," in part because of investors' suddenly renewed interest in land [31]. Foreign land deals have a long history, and some have argued that the recent development is only the latest stage in a long-term colonial or neo-colonial project [30]. But many commentators point out that the pace and scale of recent land deals are unprecedented [30,32]. This apparent change in land control raises a number of important normative questions. It would be worth reflecting further on how new this development truly is, but due to space constraints I cannot pursue that question here.

According to one estimate, in the period from 2006 to 2010 at least 70 million hectares were bought or leased by foreign investors, an area about three times the size of the United Kingdom [32–34]. About three quarters of this area were acquired for agricultural production, which includes the production of crops for biofuels. Land was also acquired for the extraction of metals and minerals, industry, tourism, and forest conversions [34,35]. Most land deals have been reported in Africa, South East

Asia, Latin America, and territories of the former Soviet Union [36]. While much of the initial media coverage of large-scale deals focused on the role of food-insecure states such as the Gulf States, private and publicly traded companies and investment firms from the EU and the US have also played an important role. EU investors have been involved in deals that cover about a quarter of the overall area involved in concluded deals [36]. Among the main drivers for the land rush are the rise in world food prices (especially the spike of 2007-8), energy policies designed to subsidize the production of biofuels, as well as an increasing interest in land as a financial asset [2,29,30].

Proponents of large-scale land deals, including the World Bank, have argued that they can contribute to sustainable development and help alleviate poverty by enhancing economic growth, facilitating technology transfer, and providing employment, all the while helping to solve energy and food crises [3,31]. This optimism is often based on the assumption that the acquired land is marginal, unused, or underutilized, making foreign investments a win-win situation. In the preface to the 2008 World Bank report *Rising global interest in farmland*, the Director of the World Bank's Agriculture and Rural Development Department, Juergen Voegele, suggested that "( . . . ) when done right, larger-scale farming can provide opportunities for poor countries with large agricultural sectors and ample endowments of land" [37], (p. XV). While the authors of the report critically identify many risks and challenges of large-scale land acquisitions, Tania Li has argued that such reports help "render land investable" by assembling complex information about arable land. They "enable a new way of thinking about 'underutilised land' as a singular thing with qualities and potentials that can be rendered commensurable according to different criteria, and made available for comparison (and investment) at continental and global scales" [31], (p. 593). Claims about "underutilization" often ignore that land may be used for a variety of land-based livelihood strategies, often under customary tenure rights [2]. Some data indicates that nearly half of all land deals target areas with existing agricultural activities [1]. As a result, land deals often lead to the dispossession and displacement of local communities [38–40]. Moreover, most large-scale projects fail to provide the promised benefits to the countries and people from which they acquired the land [2]. Large-scale land deals can lead to "food insecurity, local and global environmental damage, the loss of livelihoods, nutritional deprivation, social polarisation and political instability" [35] (p. 443).

We might conceive of these negative effects primarily as the result of ethically problematic, imprudent, or risky behavior of investor firms, the governments of host countries, local elites, and other actors. One prominent political response has been the call for codes of conduct that would regulate land deals to mitigate negative impacts and maximize the opportunities for affected communities [40,41]. But an approach that looks primarily at individual actors would leave the global structures, as well as the policies of investor countries that facilitate and promote large-scale land deals, unanalyzed. Moreover, focusing solely on the local conditions in which land deals happen would allow citizens and consumers in Northern countries to avoid reflecting on their own role in the global land rush and on the implications for their political self-understanding.

From a more global perspective, the sociologist Saskia Sassen has looked at the land rush as a symptom of larger structural transformations. In particular, she argues that territorial land governance by sovereign nations is increasingly replaced with "flow-based" forms of governance [32,42]. The latter primarily govern the flows of resources and goods in international markets, for example, through production standards in agricultural value chains, voluntary regulations in the mining sector, forest certifications, and so on [43]. As a result, what is grown on local land, for what ends, and for whom, is determined not by those inhabiting the land, but by international markets and investors. Large-scale land deals often follow an "extractive logic," as investors produce goods primarily for the global affluent, do little to improve the local economy, and leave behind environmental degradation [4,42]. Politically, they result in a "disassembling of national territory": What was once part of a national sovereign territory, in which a state or a political community exercised territorial jurisdiction (at least formally), becomes part of a material assemblage that connects local land closely to international markets and investor countries [32].

This change in land governance has significant social and political consequences–and these consequences call for philosophical reflection. More than just a change in land title, land deals change how land is used, often resulting in the eviction of complex social structures and communities. This affects the political inclusion of those who inhabit (or used to inhabit) the land, as Sassen points out: "At the extreme we might ask what is citizenship when national territory is downgraded to foreign-owned land for plantations and the rest is evicted—floras, faunas, villages, smallholders" [32] (p. 43). Host states often play a significant role in making land available for large-scale investments. For example, by creating cadasters, land records, and titles, states simplify land-based social relations and make them legible for state administration [44]. But as a result, they are likely to ignore the messy reality of existing land-based social relations and violate the land rights of local communities [45]. In Africa, for example, most land is legally defined as national or public land, though much of it is at the same time owned and used through customary regimes [46]. This raises complex questions around how to think about citizenship and citizen-state relations, in ways that might diverge from how Western political philosophers tend to understand them. This reflection would be necessary, for example, when analyzing the moral and political implications of titling programs that render some livelihoods and forms of social inclusion illegible.

In addition, land-deal induced displacements initiate new migratory flows, raising questions about the inclusion of migrants into new communities. For example, the persecution of the Rohingya in Myanmar coincided with massive land grabs for mining and agriculture by foreign investors that the country has facilitated [47]. Migration towards Europe is affected by land-grabbing in Sub-Saharan Africa, where the growing of crops for international food and biofuel markets, rather than for those who live on the land, has increased food insecurity [48–50]. Whether such migrants fit the usual political categories of refugees and "economic migrants" also calls for conceptual and normative reflection [51,52].

Given the transnational nature of the structures and processes that enable and facilitate large-scale land acquisitions, it would be a mistake to restrict our attention to local actors and communities [3]. Flow-centered governance tends to be dominated by powerful transnational actors, including Northern industry and transnational corporations [43]. Moreover, while private corporations are a central driver, the global land rush cannot be characterized simply as an erosion of public governance, as some states have played an important role in establishing, facilitating, and promoting the global economic regime in which global production networks operate. This includes trade, investments and intellectual property agreements, the deregulatory policies of the World Bank and the International Monetary Fund, the privatization of regulatory governance, the removal of worker protections, and so on [53].

Since my goal in this paper is to reflect on the responsibility of political actors, citizens, and consumers of Northern states, I will continue to pursue this point in the next section by considering the role of the European Union as a driver of the global land rush. I will assume that it is reasonable to understand the EU as an institution that represents or organizes a distinct political community (see [54] for further discussion). The EU claims the authority to make collectively binding decisions and EU citizens have standing to participate in making those decisions, for example through their national governments and through their representatives in the European Parliament. In light of the EU's role as a driver of large-scale foreign land acquisitions, I consider what the legitimate boundaries of that political community are.

## 4. The EU's Renewable Energy Directive and Transnational Governance

The European Union's Renewable Energy Directive, passed in 2009 after long discussions, establishes that 20% of all energy in the EU and 10% of all transport fuel must come from renewable sources by 2020 [55,56]. The Directive is part of the EU's efforts to combat climate change and to become "the world's number one on renewables" [57]. In general, the EU relies on a mix of governance forms [58]; its climate change policy in particular has to deal with a complex governance structure because the EU has only weak legal competences in energy and taxation policy. The Directive left a lot

of discretion to its member states on how to achieve their respective targets, and in 2014 the European Commission abandoned national renewable energy targets [59]. In the following I will abstract from these complexities and focus on how the EU manages to apply its rules externally [60]. The Renewable Energy Directive is a significant driver of large-scale foreign land acquisitions, and as a result, the EU effectively governs land far beyond its official borders.

The EU and its member states have, in effect, settled on the deployment of biofuel as one of the primary means for achieving the Directive's goal, though this is not directly mandated by the Directive. They have promoted the deployment of biofuel through, for example, tax exemptions, congestion-charge exemptions, eco-certification for Green cars, support for research and development, and so on [55]. Originally, the Commission envisaged the deployment of biofuels to be based more or less entirely on domestic agricultural production. This was motivated by a concern for energy security and the problem of agricultural overproduction in the EU [55]. But the rapid surge in demand for biofuels soon led to the emergence of global biofuel production networks [55,61]. To encourage global production, the Commission started to argue that biofuel development could boost the development of innovative farming products in developing countries, make technology transfers possible, and connect developing countries to global commodity markets and thereby facilitate rural development. It started to push for a global standardization of biofuels and included biofuels in various free trade negotiations between the EU and other regions. This situates the Directive at the intersection of a number of EU policy agendas, including access to natural resources, trade liberalization, rural development, and techno-scientific development [62]. Since conventional biofuels are produced from food-crops, the globalization of biofuel production involved large-scale land acquisitions. It is estimated that achieving the Directive's goal will require an increase of 11 million hectares in agricultural land use, an area about the size of Bulgaria [63]. Since the available arable land in the EU is not sufficient, the EU will have to outsource much of its biofuels production to the global South. The Directive is thus one of the main drivers of large-scale land acquisitions outside of the EU, as it "creates a market and thus incentives for agro-industrial biofuels development, both in the EU and in the global South" [56] (p. 690).

The environmental and social benefits of biofuels have been hotly contested. Some of the harmful effects of land use changes have already been discussed above. The Directive does disqualify some biofuels from being counted towards the target based on environmental criteria [62]. However, it did not originally address indirect land use changes, though the EU has recently reconsidered the issue [64]. Indirect land use changes occur when farmers globally respond to higher prices for food crops and convert forests, fields, and peatland into new plantations, releasing stored carbon. For example, as Germany, the EU's leading biofuel user, imported Europe-wide sources of oilseed rape, the former uses of oilseed rape had to be substituted by more palm oil from Indonesia, leading to the destruction of forest and peatland there. While indirect effects can be difficult to measure, a 2008 study by the EU's Joint Research Centre suggested that the released greenhouse gases have the potential to negate the savings from conventional biofuels. However, the EU has treated such difficulties primarily as technical issues, for example by anticipating techno-scientific innovations such as second-generation biofuels, which would make biofuels more environmentally sustainable in the future [62]. Social sustainability criteria (such as land rights of local communities or food security) were ultimately excluded from the Directive. In addition to concerns that social criteria could run afoul of global trade rules, they were excluded because of worries that the criteria would be difficult to verify [62,65]. As a result, controversies about land use change have primarily become disputes about better carbon accounting methods, for which indirect land use changes are only an "accounting error" [65] (p. 342). Knowledge about the impacts of biofuel-driven land use changes is thereby rendered "placeless." That is, from the perspective of European policy, the places affected are seen as "effectively devoid of biodiversity, of people and society, and even devoid of water" [65] (p. 347). The EU reduced its political accountability for harm to accounting for greenhouse gas emissions, mostly neglecting the profound ethical implications of land use change.

The EU's Directive shows that governments have been actively involved in facilitating the global land rush. The Directive effectively contributes to the governance of land outside of its legal jurisdiction. It does this, in part, by relying on hybrid governance forms that involve a complex configuration of public and private, domestic, transnational, and international actors [66,67]. For example, to control and verify compliance with its sustainability criteria, the EU relies on private certification schemes. These schemes can indirectly influence land governance in host countries. For example, the biofuel sector in Mozambique was developed in line with EU sustainability criteria to meet the requirements of the EU market [64]. This deep interdependence between private and public enables the EU to govern beyond its territorial borders: On the one hand, the EU needs a private certification scheme to extend its authority to apply sustainability criteria beyond its territorial borders. But on the other hand, these private certification schemes are based on incentives created by the Directive, which establish their legitimacy as market-based instruments [66]. Such hybrid forms of governance deserve more attention by philosophers, as they do not neatly fit the presumed contrast between the sovereign authority of modern states and the merely "anarchic" interactions among states and between states and transnational non-governmental actors [67–69].

The EU's ability to govern land outside of its borders is also based on bilateral investment treaties, which protect foreign investors from adverse host-state interference. By the end of 2011, there were over 3000 such treaties in force world-wide. International investment law tends to characterize land as a tradable, productive commodity that can be conceptualized in monetary terms, ignoring the complex social, political, and cultural relations in which landholding is embedded. This legal context makes inhabitants vulnerable to dispossession because legal options to defend their rights or negotiate a fair deal are limited under both national and international law [70–72]. While states may be formally free to enter such treaties, the debt crisis has eroded the negotiating power of developing countries and increased the willingness of governments to sell vast amounts of land [42,70]. In the context of this legal regime, the EU's Directive becomes effective at a distance and helps control how land outside of the EU's official borders is used.

The EU's Directive may look like a typical exercise of a political community's authority. It is designed to achieve goals that contribute to the community's common good (e.g., the reduction of greenhouse gases). As with many policies, achieving its goal requires certain trade-offs (e.g., an increased risk of food insecurity). Such trade-offs are based on judgments about the comparative value of competing goods. Legitimate political communities arguably have the authority to make such trade-offs, at least if everyone who may be affected is able to participate in the decision-making. But in this case, many of the risks and burdens of the policy fall on people who live outside of the EU's borders and who do not have any de facto standing to participate. Indeed, they are not just incidentally affected by the policy; they are subjected to authoritative rules, designed to ensure that local land use is determined by the needs of EU countries and their communities.

Given that the livelihoods of the inhabitants of land outside of the EU are governed by EU policy, we should ask whether they have moral standing to participate in the policy-making. With that, we return to the philosophical question about the legitimate boundaries of political communities that I raised in Section 2. The ability of EU citizens to continue their energy-intensive way of life depends, at least in part, on this form of governance. It seems unreasonable to regard those who make this way of life possible as outsiders. De facto, the EU is a territorially bounded political community which recognizes only long-term residents as members with the right to participate in political decision-making. But the global land rush has created transnational links that ought to be understood as political relationships, or so I argue. Citizens of wealthy EU countries have reason to consider some of those affected by the global land rush as members of their own political communities. In the remaining part of the paper, I elaborate this argument.

## 5. Social Connections and Political Responsibility

I claimed that EU citizens should see people who live on or are expelled from land that is effectively governed by EU rules as fellow members of their political community. To elaborate the argument, I will make use of Iris M. Young's *social connection model of responsibility*, which contends that our political responsibility extends to anyone to whom we are socially connected [25,73,74]. Young developed her model to describe the responsibility that consumers in the United States and Europe incur by buying products from sweatshops in developing countries. Young contrasts her account with a liability model of responsibility, which is typically applied in legal reasoning to find guilt or fault for a harm. On this latter model, it is primarily the owners and managers of the factories who are responsible for the treatment of workers and who are to be blamed for their plight. In contrast, consumers cannot be blamed because they are not sufficiently causally connected to workers and their working conditions. It is not possible to trace how each consumer's actions produce specific effects because there are too many mediating actions and events. Nonetheless, activists in the anti-sweatshop movement claim that affluent consumers in the global North share responsibility for the fate of sweatshop workers because of the structural processes that connect us to them. Young's social connection model is supposed to make sense of this claim. Her model does not require the identification of an isolatable perpetrator who directly caused bad working conditions. It does not seek to lay blame for harms that have occurred. Instead, it is forward-looking; it emphasizes our responsibility to collectively work to change structures and institutions in order to prevent further injustice.

This forward-looking responsibility, Young argues, is based on our social connection with others. When we act to pursue our own interests and goals, we do so within institutions and practices within which others act as well. Insofar as we depend on others to carry out certain tasks and form expectations on the basis of our knowledge of other people's actions, our actions assume the actions of others. Following the work of Onora O'Neill, Young argues that our ethical obligations extend to those whose actions we assume when we act because by doing so we make a practical commitment to treat them justly. In continuing to lead an energy-intensive way of life and to depend on a largely fuel-based transport infrastructure, while at the same time committing to fight climate change primarily by deploying biofuels, we objectively depend on the actions of those involved in biofuel agriculture. We also depend on people giving up, voluntarily or involuntarily, their previous land-based ways of life. If they were to resist land use changes, we would likely not be able to deploy biofuels to the extent that we planned. While it is impossible to trace the specific effects of, for example, my getting in a car to get to work, I cannot deny the social connections that an international biofuel production network creates.

Young suggests that the responsibility that emerges from social connections is political, that is, it is a shared responsibility to join or make possible collective action aimed at reforming unjust structures, something no one can do on their own. Social connections call for the creation of political institutions as a means of discharging the responsibilities that we incur in virtue of those connections. The fact that there may not currently be such institutions, or even a shared sense of mutual accountability, does not entail the lack of political responsibility. Young's account thus provides a compelling revision of the idea of a political community [75]. Political community has its foundation not in shared values, an already existing sense of solidarity, or a common will, but in shared social and economic practices that require regulation to avoid injustices. Since Young's model defines the boundaries of a political community in terms of objective social relations, rather than the will or attitudes of its members, it avoids the infinite regress of the boundary problem. It also stops the regress at the morally relevant point: our dependence on the actions of others.

Many political philosophers and political actors believe that shared nationality creates special responsibilities towards fellow nationals. They often base this on the assumption that we are more tightly connected to fellow nationals than to outsiders. More specifically, as I suggested earlier, defining political communities territorially might be justified by appealing to the shared use of territory as the basis of collective social and economic life. But as we have seen above, EU citizens in fact rely on the use of land in other countries. Only because land outside of the EU is used to grow crops

for biofuel production and some of those who depended on the land for their livelihood vacated or were expelled from it, only because of that can we use biofuel to reduce carbon emissions without making more substantive changes to our way of life. If we see the shared use of land as a proxy for connections that call for shared political institutions, then we need to recognize that it extends far beyond current nation-state borders. Young's notion of social connection does not require living side-by-side or any direct personal interaction, nor does it require a shared sense of belonging (in this respect, it differs from appeals to social membership in debates about immigration, where it is used to argue for the political inclusion of long-term non-citizens residents [24,28,76]) We may be connected primarily by complex institutions and material structures, but such connections nonetheless create political obligations. In the final section, I will defend this claim against a common objection.

## 6. Coercion, Authority, and Transnational Governance

It may be objected that social connections alone are not sufficient to ground genuinely political relationships. The social connections created by global biofuel production networks should be understood as mere interactions rather than proper associations, and thus do not create political obligations [77]. Thomas Nagel, for example, has argued that political relationships have to fulfill two additional conditions [78]. First, they have to involve coercion, in the sense that their participants have no choice but to play some role within them (see also [79]). Second, they require a kind of joint authorship, in the sense that the shared practices are conducted in the name of those who have to participate in them. In other words, political relationships require non-optional forms of cooperation that are governed by rules that come with a claim to legitimate authority. These conditions, Nagel argues, are not satisfied in the case of transnational systems of production:

> My relation of co-membership in the system of international trade with the Brazilian who grows my coffee or the Philippine worker who assembles my computer is weaker than my relation of co-membership in U.S. society with the Californian who picks my lettuce or the New Yorker who irons my shirts. ( . . . ) I doubt that the rules of international trade rise to the level of collective action needed to trigger demands for justice, even in diluted form. The relation remains essentially one of bargaining, until a leap has been made to the creation of collectively authorized sovereign authority. [78] (p. 141)

By characterizing these interactions as "bargaining," Nagel is suggesting that international trade and production networks are based on voluntary interactions between states. The rules that govern such interactions are not coercively imposed by a global institution but are the result of free negotiations. Insofar as workers in Brazil are subject to coercively imposed rules, those are the rules of a territorially bounded political community (like Brazil), but the relations between bounded communities are not coercive. Moreover, since international trade rules are the result of voluntary and self-interested bargaining, they do not involve a claim to authority that requires equal treatment in some sense.

The above discussion of the global land rush provides some material to challenge Nagel's empirical claims, at least when it comes the kinds of network discussed here. A shift to flow-based land governance leaves local residents of acquired land with little choice but to participate in a global scheme of cooperation—or else to be expelled, left without a livelihood or the ability to participate in social and economic life. Nagel might suggest that the ultimate source of such coercion is the state in whose territory the land is located, at least insofar as the state tolerates or even promotes large-scale land acquisitions by foreign investors. But as we have seen, the reality is more complicated. How the land is used, what is grown there, for whom, with what environmental and social effects, is often determined through global production networks and/or international markets. This control over foreign land is often protected by investment treaties, which limit or at least discipline the exercise of state sovereignty. Effectively, transnational companies exercise significant amounts of power over acquired land and its inhabitants, and this power is often only nominally checked by the host state. It would be wrong to suggest that host states play no role at all; we have seen that they often enable

and promote land deals. This has implications for the social and political inclusion (or exclusion) of their citizens because land governance policies that enable land deals may render certain livelihoods illegible. But this does not undermine the main claim: That the livelihoods of local residents are in many cases governed by the rules and demands of faraway communities and that they have little choice but to play by those rules.

Moreover, this form of land governance cannot be understood merely as the result of economic interactions governed by market forces. Nagel's focus on the transactions between states neglects the role of hybrid governance systems that involve both private and public actors, including states and transnational corporations, but also nongovernmental organizations (NGOs), international organizations, and so on. Complex global production networks exercise *public power*, that is, social power that pervasively effects the social conditions for the realization of individual autonomy, and thus calls for institutional regulation [68,69,80]. It is common within political philosophy and political theory to assume that all public power is concentrated within the institutional structure of sovereign states. But non-state actors such as transnational corporations and nongovernmental organizations, alongside or sometimes deeply entangled with states and intergovernmental organizations, wield public decision-making power in the form of rule-making, economic development, public service provision, and—as we have seen—land governance. In this context, the EU does not exercise its power directly and unliterally, but it is also not merely a passive beneficiary of the operations of biofuel production networks. Rather, we have seen that it facilitates their creation by using its authority to create a market for biofuels and its international power to help create a market for the land that is needed to produce them.

Global production networks involve cooperation based on relatively stable social rules. As such, they are appropriate targets for claims made by people affected by their operation to participate in the setting of those rules [80]. How such networks are to be governed legitimately is a complex and difficult question that I cannot address here (see [68] for further discussion). However, it should be noted that there are powerful non-state systems of governance that create and enforce the rules through which global production relations are facilitated. While such governance systems are usually dominated by powerful companies, they often involve the explicit agency of civil society actors, labor groups, NGOs representing smallholders, and so on, often through multi-stakeholder arrangements such as the Roundtable on Sustainable Palm Oil [66]. Thus, while there is no "joint authorship" of the kind found in sovereign democratic states, the rules that govern global production systems involve a claim to govern in the name of those who are affected [80]. To sum up, the relationships created by global biofuel production networks are embedded in coercively imposed schemes of cooperation that come with a claim to be governed in the name of those who have to participate in them, which means that they satisfy Nagel's demanding criteria.

## 7. Conclusions

In the context of the global land rush, new flows of capital have changed livelihoods and, in many cases, have led to the displacement of residents of acquired land. I explored some of the implications of these changes for how we should understand the boundaries of political communities. As a case study, I discussed the role of the European Union as a significant driver of foreign land acquisitions. Using its global power, the EU is effectively governing land far outside of its international borders and the people who live on this land or are expelled from it. This makes it possible for Europeans collectively to use biofuels to reduce carbon emissions while at the same time continuing our energy-intensive way of life. I argued that the created social connections give rise to a demand for political institutions that allow everyone who is involved to collectively regulate those connections in ways they judge most just. In other words, we ought to see those whose livelihoods are connected to (for example) global biofuel production networks as fellow members of a shared political community.

My argument is not intended as an institutional blueprint. I cannot discuss here how institutions would have to be structured to legitimately govern this political community. But the argument does

challenge the methodological nationalism that assumes the congruence of a political community, a bounded territory it inhabits, and a state that governs both. Large-scale foreign land acquisitions, as such, are not a new phenomenon. They have obvious historical precedents in the era of colonialism, and post-colonial conditions likely facilitate the more recent wave of land acquisitions [42]. But the current land rush occurs in a world of sovereign states that claim exclusive territorial control, at least formally. Assumptions of formal sovereignty and territorial control play a central role in political philosophy and theory. Many arguments for the right of states to exclude would-be immigrants, as they have been proposed by liberal political philosophers and are applied to current debates about immigration in Europe [51], are based on these assumptions. I submit that a careful look at the global land rush challenges these assumptions and raises important and complex questions for political philosophy and theory. Anyone who is interested in issues of global justice, environmental justice, global governance, global democracy, and migration, is well-advised to study the global land rush carefully.

**Funding:** The publication of this article was funded by the Qatar National Library.

**Acknowledgments:** This paper is an elaboration of a presentation given at the 2018 LANDac Conference "Land Governance and (Im)mobility: Exploring the nexus between land acquisition, displacement and migration." The author would like to thank conference participants at LANDac, the 2018 conference of the German Society for Analytic Philosophy (GAP.10), the 2018 Workshop for Political Philosophy in Berlin, the 2019 Eastern Division Meeting of the American Philosophical Association, as well as the anonymous reviewers for this journal for their helpful comments and suggestions.

**Conflicts of Interest:** The author declares no conflict of interest.

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
