# Peer review of "How Far Does the European Union Reach? Foreign Land Acquisitions and the Boundaries of Political Communities"

_land, doi:10.3390/land8030044_

Round 1

Reviewer 1 Report

This paper offers an interesting critique of liberal political philosophical positions asserting states' unilateral right to collective self-determination when assessing the ethics of closed state border practices. To defend its critique, it offers a comprehensive review of recent literature on changes occurring in land governance and their detrimental impact on the lives of peoples in the developing world. On the basis of such arguments, it points to the practical as much as philosophical importance of Young's a social connections approach to issues of responsibility and justice with some interesting case examples (bi-lateral investment agreements, the EU Renewable Energy Directive and the push towards heightened bio-fuel production globally). Overall, the paper is extremely well written and I feel offers an important contribution to debate on these issues. I would, however, suggest that the author includes a few more specific details on the position of Miller or indeed, Rawls in section two ('The Legitimate Boundaries of Political Communities') to give the non philosophy reader a greater sense of the main lines of argumentation. For instance, Miller on the need to protect the cultural priorities of peoples of state bound territories, permissable exclusivity when determining justice in relation to immigration, what are deemed 'legitimate constraints' in immigration, etc.). With this minor change, I strongly support publication of this paper.

Author Response

This paper offers an interesting critique of liberal political philosophical positions asserting states' unilateral right to collective self-determination when assessing the ethics of closed state border practices. To defend its critique, it offers a comprehensive review of recent literature on changes occurring in land governance and their detrimental impact on the lives of peoples in the developing world. On the basis of such arguments, it points to the practical as much as philosophical importance of Young's a social connections approach to issues of responsibility and justice with some interesting case examples (bi-lateral investment agreements, the EU Renewable Energy Directive and the push towards heightened bio-fuel production globally). Overall, the paper is extremely well written and I feel offers an important contribution to debate on these issues. I would, however, suggest that the author includes a few more specific details on the position of Miller or indeed, Rawls in section two ('The Legitimate Boundaries of Political Communities') to give the non philosophy reader a greater sense of the main lines of argumentation. For instance, Miller on the need to protect the cultural priorities of peoples of state bound territories, permissable exclusivity when determining justice in relation to immigration, what are deemed 'legitimate constraints' in immigration, etc.). With this minor change, I strongly support publication of this paper.

Response 1: Please provide your response for Point 1. (in red)

Thank you for the feedback. I have added a short paragraph to lay out in more detail David Miller's positions in debates about immigration and global distributive justice. 

Reviewer 2 Report

An interesting piece! Section 5 is very well written, other sections lack some clarity (flow of reading). I am missing the aspect of state-citizen relations in affected countries, without the argument seems under-complex (and becomes more vulnerable).

29 - explain 'flow based governance', e.g. in a footnote 

29 - parts of the world... 

In section 1, the term 'global land rush' is used repeatedly without defining it. This appears necessary. What are the defining features? Is this a an undisputed concept? What distinguishes the global land rush from earlier periods...? I see that an attempt is made in section 3, this is too late and also remains quite vague as it is not being distinguished clearly from earlier periods. Furthermore, given the topic of the paper, would some mentioning of neo-colonialism (and post-colonial theory?) in the context of land investments etc. not be a necessary theoretical addition at this point?

136 migrantS

146 restring? restricting

180 revise sentence structure
184 ff - can you give examples where the World Bank and the ominous 'others' make these statements? There are quite critical publications even by the WB on this issue. Not to play the devils advocate, but this is quite the broad statement and a little imprecise. It seems to serve a 'good-bad' narrative rather than help the analysis. Maybe give concrete examples of projects where the World Bank and others (define them!) support large scale land investments to make your case.

254 review sentence

255 ibid

264 grammar

 In section 4 I am missing a more general introduction on how the EU governs, in order to give some context to your analysis (e.g. Boerzel) 

324 sentence structure

Section 5: Well written and clear! I missed this clarity in previous sections. You might check if all citations in previous sections are necessary. Furthermore, outsourcing citations or side notes to footnotes might help to improve the flow of writing.

Section 6: In this (and also previous theoretical explanations) I miss the aspect of citizen-state relations especially in those countries where the land investments take place. Without this, the analysis appears incomplete (and oversimplified). Periphery-center theories and similar might help as a further explanatory factor (e.g. Wallerstein, Chase-Dunn) but there are many others that go in this direction (possibly James Scott?). 

494 sentence structure

Author Response

Point 1: An interesting piece! Section 5 is very well written, other sections lack some clarity (flow of reading). I am missing the aspect of state-citizen relations in affected countries, without the argument seems under-complex (and becomes more vulnerable).

Response 1: Thank you for your feedback and thoughtful suggestions. I have incorporated the suggestions about style, grammar, typos etc. in the text. Unfortunately, I was not able to do much about readability, since Land does not allow for footnotes, which I agree would improve the paper's flow. 

Point 2: In section 1, the term 'global land rush' is used repeatedly without defining it. This appears necessary. What are the defining features? Is this a an undisputed concept? What distinguishes the global land rush from earlier periods...? I see that an attempt is made in section 3, this is too late and also remains quite vague as it is not being distinguished clearly from earlier periods. Furthermore, given the topic of the paper, would some mentioning of neo-colonialism (and post-colonial theory?) in the context of land investments etc. not be a necessary theoretical addition at this point?

Response 2: This is a fair point. I have been using the term primarily as a proper name, without addressing some of its connotations and without providing a clear definition. Given that the term is used in much of the literature (including most surveys) without critical discussion, I think it's acceptable for me to use it, especially since I rely on what is established in the literature as the empirical basis for my normative argument. I also think that the term is preferable to more tendentious claims such as "land grabs". 

But I recognize the conceptual vagueness of the term and some of the questions it raises (questions about similarities with earlier periods and questions about the connection of current land deals with colonial and post-colonial land acquisitions). I address some of these questions in the revised first paragraph of section 3. I took out most uses of the term before section 3 and substituted them with more neutral descriptions –– I didn't want to clutter the introduction with more asides. 

The question of neo-colonialism is an important one, but I couldn't see any way to address the issue without adding another section. I now briefly mention the issue at 265. 

Point 3: 184 ff - can you give examples where the World Bank and the ominous 'others' make these statements? There are quite critical publications even by the WB on this issue. Not to play the devils advocate, but this is quite the broad statement and a little imprecise. It seems to serve a 'good-bad' narrative rather than help the analysis. Maybe give concrete examples of projects where the World Bank and others (define them!) support large scale land investments to make your case.

Response 3: Fair point. I have added a quote from the preface of the Rising Global Interest in Farmland report, while at the same noting the more critical voices of its authors. I also added a brief discussion of Tania Li's argument about the role such reports play in making land investible to add some nuance.

Point 4: Section 5: Well written and clear! I missed this clarity in previous sections. You might check if all citations in previous sections are necessary. Furthermore, outsourcing citations or side notes to footnotes might help to improve the flow of writing.

Response 4: See response 1 above. 

Point 5: Section 6: In this (and also previous theoretical explanations) I miss the aspect of citizen-state relations especially in those countries where the land investments take place. Without this, the analysis appears incomplete (and oversimplified). Periphery-center theories and similar might help as a further explanatory factor (e.g. Wallerstein, Chase-Dunn) but there are many others that go in this direction (possibly James Scott?). 

Response 5: I added a brief discussion of what I understood the reviewer's point to be at 366 (in section 3). I agree that a less simplified argument needs to address the relationship between citizen and state in countries where land investments take place. This is another point that would require at least another section to address fully, but I hope that the added passage provides at least some indication where that argument would go. I also included a short reference in section 6 at 520. 

Thanks again for the thoughtful and constructive comments! 

Reviewer 3 Report

The  study  is more descriptive and requires more strong scientific support in order to strength the discusion and conclusions.

It  is necessary to show  similarities and contrasts in relation with other real conditions around the world cases.

it  is necessary to support the research with more discussion and evidences in the international level supporting the conclusions.

The author needs to discuss the real impacts of this kind of actions  with criteria of sustainable development

Additionally, I think the author must give more details how  identified  the study problem and justification. In addition, he needs to discuss  what it would happen if the problem is not attended:  social, ecological, economics, legal and environmental policy impacts.  In addition, it is necessary to include multidimensional strategies to  guarantee the long-term functioning of the system in order to achieve  sustainable development.

Author Response

Point 1: The  study  is more descriptive and requires more strong scientific support in order to strength the discusion and conclusions.

Point 2: It  is necessary to show  similarities and contrasts in relation with other real conditions around the world cases.

Point 3: it  is necessary to support the research with more discussion and evidences in the international level supporting the conclusions.

Point 4: The author needs to discuss the real impacts of this kind of actions  with criteria of sustainable development

Point 5: Additionally, I think the author must give more details how  identified  the study problem and justification. In addition, he needs to discuss  what it would happen if the problem is not attended:  social, ecological, economics, legal and environmental policy impacts.  In addition, it is necessary to include multidimensional strategies to  guarantee the long-term functioning of the system in order to achieve  sustainable development.

Response: 

Thank you for taking the time to read my manuscript, I very much appreciate it.

Unfortunately, I found it difficult to address the very general suggestion to provide more scientific support (points 1 and 3). If you are so inclined, I would appreciate somewhat more specific suggestions about which parts of the argument need more support, or why the cited literature does not provide sufficient support. 

That being said, I suspect that there may be a misunderstanding about the aim of my paper. My paper is not explanatory. Instead, as a political philosopher, I discuss a normative/moral question about the legitimate boundaries of political communities. I motivate and elaborate this question in detail in section 2. In the "more descriptive" parts of the paper (sections 3 and 4), I summarize and reflect on what I take to be relatively established facts about the global land rush (based on the literature). This is supposed to serve as the empirical starting point for the normative argument that I develop. But I am not aiming to contribute new insights about the causes, functioning, or effects of foreign land acquisitions. My contribution in the paper is normative, not explanatory or descriptive: I develop an argument to support the claim that (as a result of the global land rush and the role EU policy plays in it) the political community that EU citizens belong to reaches far beyond the EU’s official borders.

This means that the questions raised in point 2, 4, and 5 are beyond the scope of the paper (and, frankly, my expertise). The paper does not try to compare the global land rush to other conditions, discuss the impact of land acquisition  (beyond what is already established in the literature) , or consider how sustainable development can be achieved. I do not address sustainable development in any detail, not because that's not a worthwhile question, but because I focus on the political responsibility of citizens of EU countries for the effects of the global land rush (I discuss this point at 69 ff. in the paper). 

Reviewer 4 Report

This is a very well-written piece that should be of great interest to the readers of Land. The author lays out their argument very clearly and with a strong basis in the literature. My only suggested changes are minor copy-editing issues, detailed below. Overall, I found this to be a fascinating read and very well-done.

Detailed suggestions:

Page 3, line 125: The use of “etc., etc.” is probably not appropriate in an academic manuscript. I would suggest removing or editing.

Page 4, line 146: “restring” should be “restricting”

Page 5, line 199: “conducts” should be “conduct”

Page 5, line 203: “a focusing” should be “focusing”

Page 5, line 207: “for” should be “of”

Page 7, line 311: “scheme” should be “schemes”

Page 7, line 321: “that landholding is embedded in” should be “in which landholding is embedded”

Page 7, line 324: “to” should be “the”

Page 8, line 370: “acts” should be “act”

Page 8, line 374: “intense” should be “intensive”

Page 9, line 417: “chance” should be “choice”

Page 10, line 494: “a nation-state govern both” should be “a nation-state that govern both”

Author Response

This is a very well-written piece that should be of great interest to the readers of Land. The author lays out their argument very clearly and with a strong basis in the literature. My only suggested changes are minor copy-editing issues, detailed below. Overall, I found this to be a fascinating read and very well-done.

Detailed suggestions:

Response: Thank you. I addressed the suggested changes in the revised version. 

Round 2

Reviewer 3 Report

I have revised the new version. The manuscript has been
significally improved, therefore, I agree with its publication of

the article.
 Thanks again.